# Polymer Ring–Flexure–Membrane Suspended Gate FET Gas Sensor: Design, Modelling and Simulation

**DOI:** 10.3390/mi14050944

**Published:** 2023-04-26

**Authors:** Joel Zacharias, Pramod Martha, V. Seena

**Affiliations:** Department of Avionics, Indian Institute of Space Science and Technology (IIST), Thiruvananthapuram 695547, India

**Keywords:** nanomechanical sensor, polymer MEMS, CMOS-MEMS, RFM-SGFET, hydrogen sensor

## Abstract

This work reports the design, modelling, and simulation of a novel polymer MEMS gas sensor platform called a ring–flexure–membrane (RFM) suspended gate field effect transistor (SGFET). The sensor consists of a suspended polymer (SU-8) MEMS based RFM structure holding the gate of the SGFET with the gas sensing layer on top of the outer ring. During gas adsorption, the polymer ring–flexure–membrane architecture ensures a constant gate capacitance change throughout the gate area of the SGFET. This leads to efficient transduction of the gas adsorption-induced nanomechanical motion input to the change in the output current of the SGFET, thus improving the sensitivity. The sensor performance has been evaluated for sensing hydrogen gas using the finite element method (FEM) and TCAD simulation tools. The MEMS design and simulation of the RFM structure is carried out using CoventorWare 10.3, and the design, modelling, and simulation of the SGFET array is carried out using the Synopsis Sentaurus TCAD. A differential amplifier circuit using RFM-SGFET is designed and simulated in Cadence Virtuoso using the lookup table (LUT) of the RFM-SGFET. The differential amplifier exhibits a sensitivity of 2.8 mV/MPa for a gate bias of 3 V and a maximum detection range of up to 1% hydrogen gas concentration. This work also presents a detailed fabrication process integration plan to realize the RFM-SGFET sensor using a tailored self-aligned CMOS process adopting the surface micromachining process.

## 1. Introduction

In recent years, there has been a growing demand for gas sensors in the chemical, aerospace, and biomedical industries to detect gases, volatile organic compounds (VOCs), and biological materials. These applications require compact and affordable gas sensors with high sensitivity, high selectivity, and fast response time. MEMS gas sensors are suitable to meet this growing demand due to their small size, low power consumption, low fabrication cost, and high sensitivity [1,2]. Some commonly used transduction methods in MEMS gas sensors include optical, piezoelectric, piezoresistive, and capacitive methods [3,4,5,6]. Each of these methods has its advantages and limitations. Thus a suitable transduction method needs to be selected based on the requirement.

Among the different types of MEMS gas sensors, the nanomechanical cantilever (NMC) gas sensors have been widely studied due to their low cost, ability to detect minute amounts of the target materials, and real-time detection [7,8,9,10]. Optical transduction is commonly used in NMCs due to their high sensitivity; however, they are bulky, cannot be applied in a multi-dimensional array, and require time-consuming laser alignment, and it is challenging to get readouts on opaque liquids [11,12,13,14,15]. The limitations of optical transduction can be overcome by using NMCs integrated with electrical transduction, such as piezoresistive and capacitive schemes. NMCs with integrated piezoresistors do not require complex and bulky laser alignments and hence can be deployed to a multi-dimensional array and can be used to detect any liquids or gaseous analytes [16,17]. However, the sensitivity is far below the optical NMCs, limiting their application as gas sensors. One of the methods to enhance the sensitivity of piezoresistive NMCs is to use polymers such as SU-8 as the structural layer material. High sensitivity polymer-based NMC sensors with high gauge factor materials such as SU-8/Indium tin oxide (ITO), SU-8/carbon black (CB), and SU-8/carbon nanotubes (CNT) have been reported [18,19,20]. Other methods involve structural modifications in NMC, which reduce the area available for the analyte or the gas molecules to interact with the sensing layer [21]. The membrane-type surface stress sensor (MSS) is an alternate membrane-based gas sensor that consists of an adsorbate membrane supported by four constricted sensor beams on which the piezoresistors are embedded [22,23]. Silicon-based MSS has demonstrated sensitivity that is 20 times higher than conventional NMCs. Based on MSS geometry, the nanomechanical membrane flexure (NMF) sensor using silicon and SU-8 with high gauge factor ITO as piezoresistor has been reported [24,25]. However, the sensitivity of these membrane-based sensors is still below that of optical NMCs.

The capacitive transduction method offers low-temperature drift, low noise, high sensitivity, and low power dissipation without needing any functional materials. A silicon-based capacitive NMC for detecting VOCs has been reported, where the detection is based on the capacitance changes due to the induced surface stress [26]. An ultra-low power capacitive MEMS hydrogen sensor based on aluminum–palladium bimorph configured as a clamped–clamped beam has been reported with high sensitivity and fast response time [27]. The NMC and membrane-based capacitive sensors have a limitation due to the deformation of the electrode during sensing, leading to a non-uniform change in the capacitance along the electrode area. A novel MEMS capacitive hydrogen sensor with an inverted T-shaped electrode and ring-shaped palladium has been reported in which the electrodes are always moving in parallel, leading to high capacitance change [28,29]. The sensitivity is reported to be three times higher than conventional sensors. For capacitive transduction, the sensitivity is proportional to the sensor’s capacitive value, which implies the need for a large area design. Thus its main shortcomings are the large area requirement and the need for low noise capacitance-to-voltage conversion.

The main drawback of these commonly used transduction schemes is that they are passive elements, resulting in a low output signal-to-noise ratio (SNR). The piezoresistive sensors require a Wheatstone bridge followed by a signal conditioning circuit to enhance the output SNR [18]. Similarly, the capacitive sensors require capacitance-to-voltage converters followed by a signal conditioning circuit [30]. This shortcoming can be mitigated using an active field effect transistor (FET) based transduction. The inherent amplification property of a FET improves the output SNR and allows the easy integration of the sensor with the readouts circuits [31]. There are several reported studies where MEMS sensor technology is coupled with suspended gate field effect transistors (SGFET) for applications such as accelerometers, pressure sensors, and biosensors [32,33,34,35,36]. In such sensors, the mechanical input causes a displacement of the suspended gate. This leads to a change in gate capacitance and thus the drain current of the SGFET. Such suspended gate FET sensors are fundamentally an active transduction counterpart of capacitive sensing, but with a potential for higher SNR as compared to capacitive sensors. Integrating SGFET based transduction element with an appropriate MEMS structure design can lead to the development of a high sensitivity MEMS-FET gas sensing platform.

We propose a novel MEMS sensor design that can meet the requirements and challenges posed by the growing demand for MEMS gas sensors. This gas sensor is an integrated MEMS-FET sensor called a ring–flexure–membrane suspended gate field effect transistor (RFM-SGFET) sensor. The sensor consists of a suspended ring–flexure–membrane (RFM) holding the gate of the FET and a fingered FET configuration. The gas sensing layer is located on top of the outer ring of the RFM-SGFET. As proof of concept for gas sensing applications, the sensor is designed and simulated for sensing hydrogen gas. The use of polymer structural layer SU-8 enhances the sensitivity of the sensor and eases the fabrication process. The novel sensor design ensures efficient transduction of gas adsorption-induced nanomechanical motion input to change the output current of the FET, thus improving the sensitivity and ensuring a high out SNR and easy integration to the electronic circuit.

## 2. RFM-SGFET Sensor Design and Operation

The sensor consists of a SU-8 based ring–flexure–membrane architecture as shown in Figure 1a. An octagon-shaped suspended ring is connected to a circular membrane using four flexures. The sensing layer is located on top of the outer ring. The suspended gate electrode is located underneath the circular membrane (Figure 1b), and there is an air gap between the gate and the channel. The SGFET consists of a fingered FET arrays that are located on the substrate below the membrane.

The operation of the RFM-SGFET sensor is shown in Figure 1b. During gas adsorption, compressive stress is induced in the sensing layer, which causes the deformation of the ring and flexure, leading to a downward displacement of the membrane. This gate displacement leads to an increase in gate capacitance of the SGFET, thus causing a change in the output drain current. Compared to conventional membrane sensors with a sensing layer spanning the entire membrane, the ring-shaped sensing layer in this design ensures that the membrane always displaces downwards during gas adsorption. Moreover, the low stiffness flexures ensure that the membrane does not deform or bend during displacement, leading to a constant change in the gate capacitance throughout the gate area. This provides a considerably more significant change in capacitance than conventional membrane sensors. The detailed design specifications are arrived at based on the MEMS and TCAD simulations of this device.

## 3. Simulation and Modelling

The sensor required a multi-domain approach, as it involves mechanical element (MEMS), semiconductor physics (FET), and circuit systems. The simulation methodology is shown in the flow chart in Figure 2 [37,38].

The MEMS RFM structure is modeled as a capacitor and simulated using the FEM software CoventorWare 10.3. The MemMech module is used to analyze the mechanical behavior of the membrane during hydrogen adsorption. The CoSolve module is used to analyze the capacitance change and pull-in voltage. The fingered SGFET is designed and simulated using Synopsis Sentaurus TCAD. The change in the air gap obtained from the mechanical analysis is used to analyze the change in the drain current and the threshold voltage of the SGFET.

### 3.1. MEMS Ring–Flexure–Membrane Design, Simulation, and Electromechanical Analysis

The primary design consideration for our sensor is the uniform change in gate capacitance during hydrogen adsorption. The conventional membrane-based MEMS capacitive gas sensor design is unsuitable, as the membrane and the electrode deforms with a curvature during the gas adsorption, leading to a non-uniform capacitance change throughout the electrode area. The deformation of an octagon membrane and a ring–flexure–membrane structure, both with a ring-shaped palladium sensing layer under compression, is shown in Figure 3a. A non-uniform deformation is observed in the octagon membrane, whereas in the RFM structure, a constant downward displacement is observed throughout the membrane area. It is very evident that this is attributed to the connecting flexures with very low stiffness in the RFM structure, leading to constant downward movement of the central membrane. Hence, a novel ring–flexure–membrane design was considered with very low stiffness flexures holding the membrane. This design is a further improvement of our previously reported work on capacitive sensors [30]. Figure 3b shows the deformation of the membrane for different shapes of the sensing palladium film under compression. It is observed that only the ring-shaped palladium ensures a parallel downward displacement of the membrane during hydrogen adsorption. The ring-shaped palladium, along with the low stiffness flexures, ensure that the membrane does not deform or bend. SU-8 is used as a structural layer, and its thickness is chosen as 7 μm based on the SU-8 formulations that can be deposited at our facility. The outer and inner diameters of the octagon ring are 450 μm and 375 μm, respectively. The diameter of the membrane is 300 μm to ensure that an array of FET devices can be accommodated under it to act as an SGFET array. Due to the area constraint between the ring and the membrane, the flexure width is fixed at 15 μm. The length of the flexure is varied from 204 μm to 102 μm in the simulation. The suspended gate of the FET is simulated as a capacitor and consists of an array of 17 rectangular movable and fixed electrodes with length 10 μm and width 210 μm. The gate length of 10 μm corresponds to the channel length of the SGFET. A channel length of 10 μm is considered, as the pseudo-short channel effects in the SGFET become severe below 10 μm due to poor gate control [33]. The air gap between the fixed and movable electrodes is 2 μm. To simulate the effect of hydrogen adsorption, an equivalent value of compressive stress is applied on the palladium ring. The initial stress on the palladium film is assumed to be zero. The sides of the outer ring are fixed in the simulations.

For the mechanical and electromechanical simulations, the applied stress on the palladium ring ranges from −100 MPa to −900 MPa. Based on the previously reported experimental values of stress induced by hydrogen gas on palladium, we have selected this stress range because it corresponds to less than 1% hydrogen gas concentration [39]. The mechanical and electromechanical characteristics of the proposed sensor are shown in Figure 4. The displacement profile for a sensor with a flexure length of 204 μm is shown in Figure 4a. The membrane does not deform or bend during gas adsorption. Hence, the RFM configuration helps achieve uniform capacitance change throughout the gate area. A parametric analysis was carried out for the flexure length so as to meet the aforementioned designed constraints and uniform displacement of the membrane. The parametric analysis of membrane displacement for flexure length from 204 μm to 102 μm is shown in Figure 4b. For constant stress, the membrane displacement decreases as the flexure length is reduced, whereas the membrane does not deform for this range of flexure lengths. The membrane tends to deform with applied stress for flexure lengths below 102 μm. Hence, they are not considered in the parametric analysis discussed here.

A parametric study using electromechanical analysis was carried out to analyze the variation in pull-in voltage and the gate capacitance for different gas concentrations and different flexure lengths. The analysis of the pull-in voltage is required to determine the operating gate bias and the range of detection of the sensor. The pull-in voltage plot for different flexure lengths with varying stress values is shown in Figure 4c. The correlation of flexure length to the range of detection can be observed, as the sensors with shorter flexure lengths have higher pull-in voltage and vice-versa. The capacitance plot with varying stress for different flexure lengths is shown in Figure 4d. For a gate voltage of 3 V, the capacitance change observed is higher as compared to previously reported MEMS capacitive hydrogen sensors [29]. The sensitivity also depends on flexure length, as the sensor with larger flexure length exhibits larger capacitance change, hence higher sensitivity. However, the range of detection is limited due to the pull-in effect. The sensor can be optimized to detect up to 1% hydrogen concentration based on the reported experimental values of stress induced on palladium for different hydrogen gas concentrations [39].

A parametric study was carried out to analyze the change in the displacement of the membrane when the area of the ring-shaped sensing layer is varied and is shown in Figure 4e. The palladium ring inner diameter was varied from 375 μm to 300 μm, and the membrane diameter was varied from 300 μm to 325 μm. It is observed that the as the inner diameter of the palladium ring is decreased, the membrane displacement increases, resulting in higher sensitivity. This is due to the increase in the area of the palladium ring with the decrease in the inner diameter, resulting in a larger area of palladium under compression during hydrogen adsorption.

The MEMS simulations have clearly demonstrated that the RFM architecture ensures a uniform change in gate capacitance throughout the gate area during hydrogen adsorption. The sensor with a flexure length of 204 μm showed the maximum sensitivity, whereas the sensor with 102 μm flexure length showed the maximum range of detection. The parametric analysis demonstrates the correlation of sensitivity and the detection range on the flexure length and the area of the sensing layer. Thus, the RFM sensor design is a versatile platform that allows the user to optimize the performance parameters such as sensitivity and the range of detection according to requirement.

### 3.2. Analytical Model of the Fingered SGFET

In the RFM-SGFET sensor, the suspended gate of the FET displaces downwards due to gas adsorption, and the air gap between the gate and the channel changes. This causes a change in the gate capacitance, leading to a change in the output drain current. Hence, the stress induced in the MEMS structure due to the gas adsorption makes the gate move downwards from its initial position and varies the drain current. The drain current is modulated by the effective gate capacitance per unit area, which is defined as
(1)$$C_{eff}=\frac{C_{ox}C_{air}}{C_{ox}+C_{air}}$$
where $C_{ox}=\frac{\epsilon_{ox}}{t_{ox}}$. Here, $\epsilon_{rox}$ (= 3.9) is the relative dielectric permittivity the of gate oxide and $\epsilon_{0}$ is the permittivity of free space. The threshold voltage varies with the $C_{eff}$ and is given as
(2)$$V_{th}=\phi_{MS}+2\psi_{F}-\frac{Q_{D}}{C_{eff}}-\frac{Q_{f}}{C_{eff}}$$
where $\psi_{F}=\frac{kT}{q}ln\left(\frac{N_{a}}{n_{i}}\right),$ and $Q_{D}=-\sqrt{4\epsilon_{Si}qN_{A}\psi_{F}}.$ Here, $\phi_{MS}$, $Q_{f}$, and $n_{i}$ are the work function difference, fixed oxide charge, and intrinsic carrier concentration ($1.45\;\times \;10^{10}$ cm${}^{-3}$), respectively.

The drain current ($I_{D}$) in saturation mode varies with both threshold voltage ($V_{TH}$) and effective gate capacitance ($C_{eff}$) as
(3)$$I_{D}=\frac{\mu_{n}C_{eff}W}{2L}{\left(V_{GS}-V_{th}\right)}^{2}$$The suspended gate FET sensitivity is defined as the change in $I_{D}$ with respect to to a change in *x*, i.e., $dI_{D}/dx$; *x* is the air gap between the gate and the channel. The sensitivity is given as [32]
(4)$$S=\frac{dI_{D}}{dx}=p-\frac{q}{{\left(x+\alpha \right)}^{2}}$$
where $p=\lambda \delta^{2}$ and $q=\lambda {\left(V_{GS}-\beta \right)}^{2}$ are constants for a given $V_{GS}$. The parameters that decide *p* and *q* are β = $\phi_{ms}$+2$\phi_{F}$, δ = −$\frac{Q_{D}+Q_{f}}{C_{eff}}$, λ = $\frac{\mu_{n}\epsilon_{0}W}{L}$, and α = $\frac{t_{ox}}{3.9}$. The fingered technique is used to improve the sensitivity of the gas sensor by ‘n’ times where n is the number of fingers. The value of the n depends on the electromechanical analysis of the MEMS structure.

### 3.3. TCAD Design and Simulation of the Fingered SGFET Array

The design and analysis of the SGFET array are carried out in Synopsys Sentaurus TCAD Process tool by following tailored self-aligned CMOS process flow. The detailed process flow, illustrated in Figure 5, is as follows. The SGFET design parameters are given in Table 1.

The starting substrate is a Boron-doped (10^15^ cm^−3^) silicon.

The gate dielectric is formed by thermal oxidation;The 200 nm thick polysilicon is deposited and patterned to define the gate area;Lightly doped source and drain are obtained by ion implantation (arsenic (As), 2.7 × 10^14^ cm^−2^, 10 keV);Deposition and patterning of nitride spacer (Si_3_N_4_);Deep n+ source and drain are obtained by ion implantation (As, 5.14 × 10^15^ cm^−2^, 65 keV);The spacer nitride and polysilicon are removed from the substrate;The second part of the gate dielectric is the air gap, which, in TCAD, is defined by a user-defined material ’air gap’ with all the properties of air included in the material database;Aluminum (Al) metal is deposited and patterned to define gate and contact for source and drain.

The change in the air gap obtained from MEMS simulations was used to analyze the change in the drain current of the SGFET. The designed SGFET using the above process flow is shown in Figure 5h. The output characteristics of an array of 17 fingered SGFET for the membrane with a flexure length of 204 μm and a gate voltage of 3 V for varying stress values is shown in Figure 6a. An increase in the drain current is observed with the increase in the compressive stress on the palladium film. This is due to the reduced air gap between the gate and channel when the hydrogen gas concentration increases. The parametric study of the variation in the output drain current for different flexure lengths is shown in Figure 6b. For a constant gate voltage and applied stress, it is observed that the drain current decreases with the decrease in the flexure length; hence, the sensor’s sensitivity decreases when the beam length is reduced. These results are in agreement with the mechanical simulations, where it is observed that the membrane displacement decreases with decreases in the flexure length.

The threshold voltage for different flexure lengths with varying stress values is shown in Figure 6c. The threshold voltage decreases with the increase in applied stress. This is due to the decrease in air gap with the increase in hydrogen concentration. When the flexure length is reduced, the threshold voltage increases. For effective operation of the sensor, the operating gate voltage must be above the threshold voltage and below the pull-in voltage for the given range of detection. On comparing the threshold voltages in Figure 6c with the pull-in voltage plot in Figure 4c, it is observed that the threshold voltage values are well below the pull-in voltage. Thus, the FET transfer characteristics (threshold voltage range) and the electromechanical pull-in analysis (pull-in voltage) for varying hydrogen gas concentrations can be used to obtain the operating voltage and the range of detection of the sensor. The analytical and simulation results for SGFET sensitivity are plotted in Figure 6d. The constant parameters given in Equation (Equation 4) are found to be *p* = 2.7 C^2^ m^3^ V^−1^ s^−1^ F^−1^ and *q* = 10.6 V·F m^−1^ s^−1^.

### 3.4. RFM-SGFET Differential Amplifier

The change in environmental parameters such as temperature and humidity can cause a change in the performance parameters of the sensor. To mitigate the environmental effects on the sensor performance, a differential amplifier configuration of the RFM-SGFET sensor is considered. The differential amplifier with variable and fixed air gap RFM-SGFETs is designed using Cadence Virtuoso circuit simulation platform. For this, the RFM-SGFET sensor with a flexure length of 102 μm, which showed the maximum range of detection, is used. In the differential pair, only one of the RFM-SGFETs has a palladium thin film ring leading to deformation with incoming hydrogen gas, thereby varying the air gap of SGFET. The reference SGFET without any palladium thin film coating does not respond to the hydrogen gas and hence maintains a constant air gap throughout the operation.

A resistive load common source differential amplifier is designed in Cadence Virtuoso. The differential RFM-SGFET hydrogen sensor circuit is shown in Figure 7a. The look-up-table (LUT) approach is followed to incorporate the RFM-SGFET characteristics in circuit design. The look-up tables are generated by incorporating the results obtained from FEM and TCAD simulations for different stress values and operating voltage points. Using Verilog-A, these LUTs are added to the circuit. The input to the MEMS LUT is a voltage signal ($V_{\sigma }$), which is the electrical representation of stress values for different hydrogen gas concentrations, and the output is the corresponding air-gap values (*x*). The two inputs to the SGFET are air gap (*x*) and gate voltage ($V_{GS}$). The initial air gap is 2 μm in both SGFETs. The drain current ($I_{Dl}$) is varied with the displacement of the membrane due to the adsorption of hydrogen gas. However, $I_{Dr}$ is not changing due to the constant air gap maintained by the reference device. The difference between $I_{Dl}$ and $I_{Dr}$ is read at the differential amplifier’s output node, which is fed into an amplifier with a gain ($A_{V}$) of 15. The load resistor ($R_{L}$) is 5 kΩ, and the current source value is 50 μA. The $R_{L}$ and $A_{V}$ values are chosen so that output will never cross the upper limit of the supply voltage ($V_{DD}$ = 3.3 V).

The circuit response to different stress signals of the differential amplifier is shown in Figure 7b and the sensitivity is found to be 2.8 mV/MPa. The output voltage of the differential amplifier for 0.5% and 1% hydrogen concentrations are highlighted in Figure 7b. The sensor circuit response is 2.71 V for 1% hydrogen concentration. The device design discussed here is a proof of concept with the application in hydrogen gas sensing with concentrations up to 1%. Though the extracted sensitivity is for a particular dimensions of this design with a capability to detect hydrogen concentrations in the sub-ppm range, the RFM sensor architecture with FET provides versatility in tuning the detection limits to a wide range of gas concentrations. This could be achieved by varying the Pd ring, flexure stiffness, and the area of the gate electrodes.

## 4. Fabrication Integration Plan

The fabrication process integration for the realization of RFM-SGFET sensor is illustrated in Figure 8. The process integration plan is based on low-temperature polymer MEMS on a tailored CMOS process.

The sensor fabrication process begins with the fabrication of the n-channel SGFET using a self-aligned CMOS fabrication method. The fabrication process of the fingered n-channel SGFET is similar to the fabrication process shown in Figure 5a–f.Aluminum is then sputtered to a thickness of 50 nm and patterned to form the source and drain electrode.LOR-20B of 2 μm thickness is spin-coated on the substrate and patterned by wet etching. This will act as the sacrificial layer for the RFM structure.The suspended gate is then deposited by sputtering 250 nm of aluminum, and then it is patterned by wet etching.The SU-8 based RFM layer of 7 μm thickness is then spin-coated and patterned by wet etching.The sensing layer can be deposited onto the SU-8 membrane by lift-off deposition of palladium using sputtering.The RFM structure is then released by the removal of the sacrificial layer by wet etching. This is followed by drying the substrate in a critical point dryer to avoid stiction.

## 5. Conclusions

The challenges associated with the growing demand for gas sensors and the limitations of the commonly used MEMS gas sensors can be addressed by our proposed RFM-SGFET gas sensor platform. This integrated MEMS-CMOS device has been designed, simulated, and implemented as a hydrogen gas sensor. During hydrogen adsorption, the novel MEMS RFM architecture ensures a constant change in the gate capacitance throughout the gate area, causing a change in the output drain current of the SGFET. The parametric analysis of the sensor demonstrates the versatility of the sensor platform, as the sensitivity and the range of detection can be tuned based on user requirements. The sensor fabrication is made simpler and cost-effective through the use of SU-8 as a MEMS structural layer. The system level implementation of the RFM-SGFET sensor in a differential amplifier circuit shows a sensitivity of 2.8 mV/MPa. Our sensor offers new possibilities for developing compact, affordable, high-sensitivity MEMS sensors with a design compatible with low thermal budget post-CMOS-MEMS integration.

## Figures and Tables

**Figure 1 micromachines-14-00944-f001:**
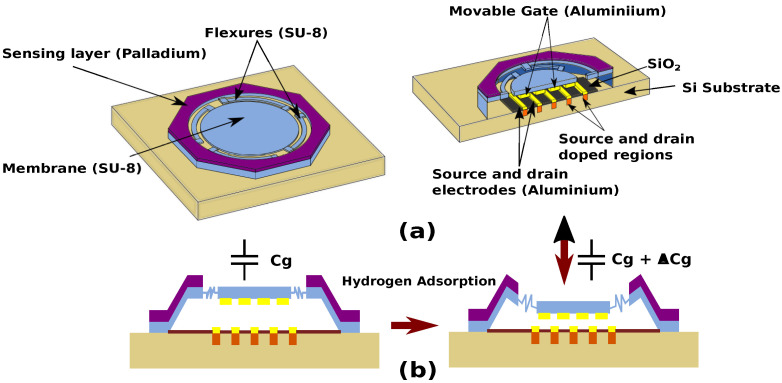
(**a**) Schematic and (**b**) operation of the ring–flexure–membrane sensor.

**Figure 2 micromachines-14-00944-f002:**
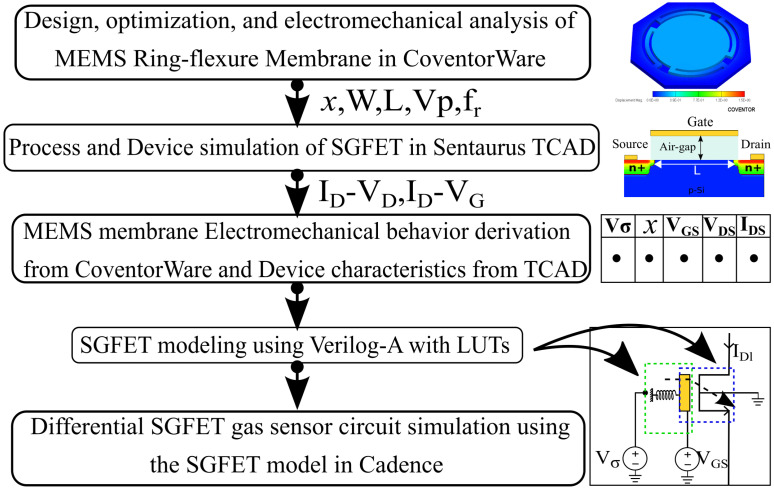
Flow chart of the simulation methodology.

**Figure 3 micromachines-14-00944-f003:**
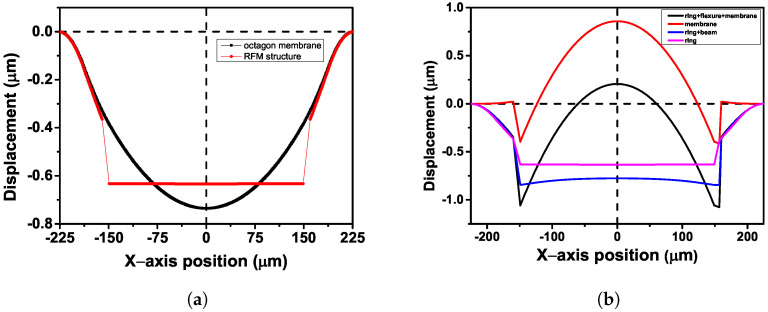
(**a**) Displacement profile of an octagon membrane and a RFM structure and (**b**) effect of the shape of the sensing palladium film on the membrane deformation.

**Figure 4 micromachines-14-00944-f004:**
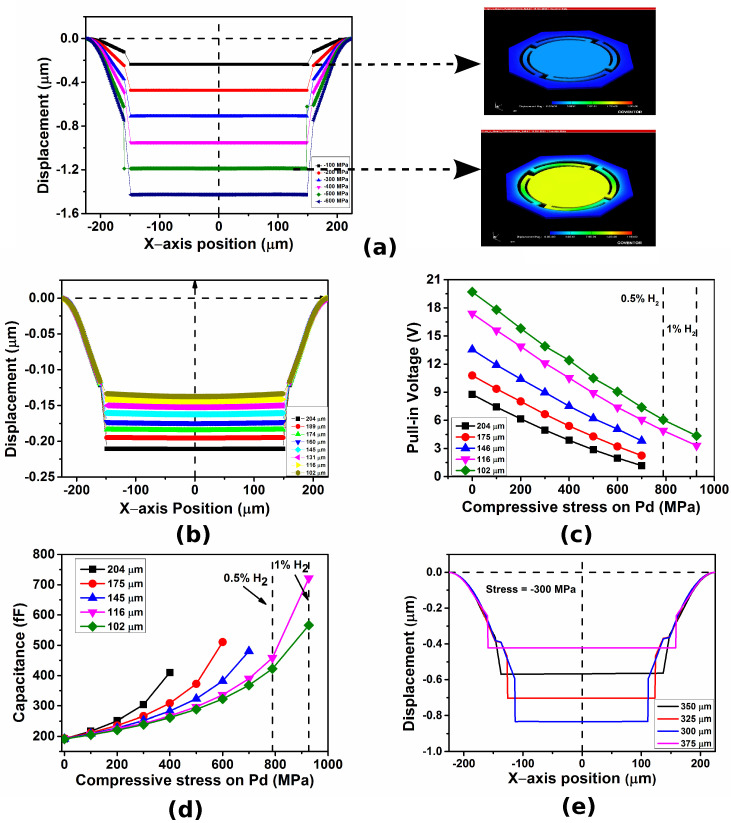
(**a**) Displacement profile for a sensor with flexure length of 204 μm, (**b**) displacement profile for varying flexure lengths, (**c**) pull-in voltage for different flexure lengths with varying stress, (**d**) capacitance for different flexure lengths with varying stress, and (**e**) displacement profile for varying areas of the ring-shaped sensing layer.

**Figure 5 micromachines-14-00944-f005:**
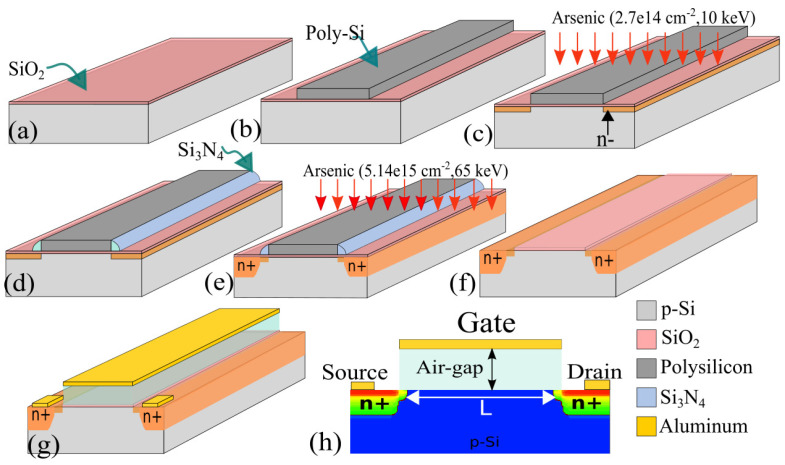
TCAD process flow of 17-fingered SGFET: (**a**) thermal oxidation to get gate dielectric (SiO${}_{2}$), (**b**) polysilicon (200 nm) deposition and patterning, (**c**) ion implantation for lightly doped drain (LDD) to minimize short channel effects, (**d**) spacer nitride (Si${}_{3}$N${}_{4}$), (**e**) ion implantation for deep n+ source and drain, (**f**) removal of polysilicon and nitride, (**g**) defining 1500 nm air gap followed by aluminum metalization, (**h**) designed SGFET with cross-sectional views and doping concentration.

**Figure 6 micromachines-14-00944-f006:**
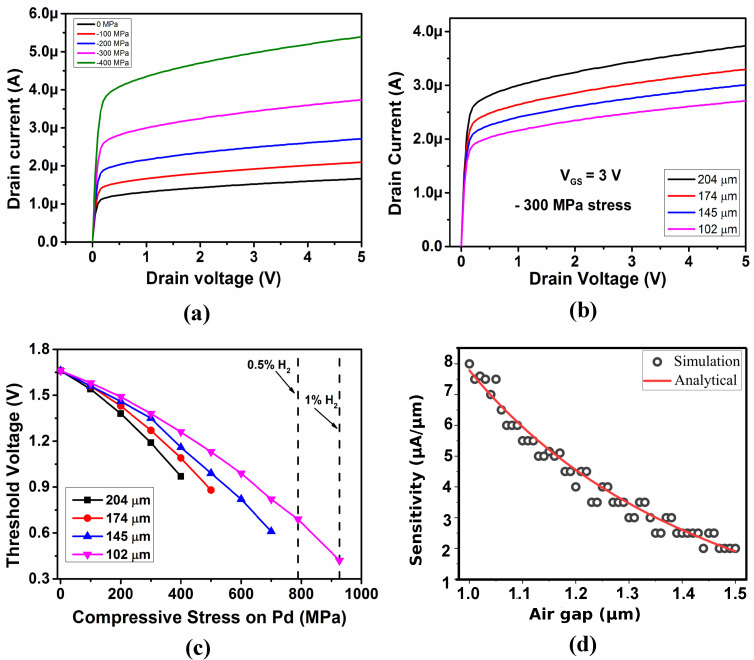
(**a**) Output characteristics of the RFM–SGFET with flexure length of 204 μm with varying stress, (**b**) output characteristics of the RFM–SGFET for different flexure length and constant stress, (**c**) threshold voltage values for different flexure length with varying stress, and (**d**) sensitivity of SGFET with respect to the air gap.

**Figure 7 micromachines-14-00944-f007:**
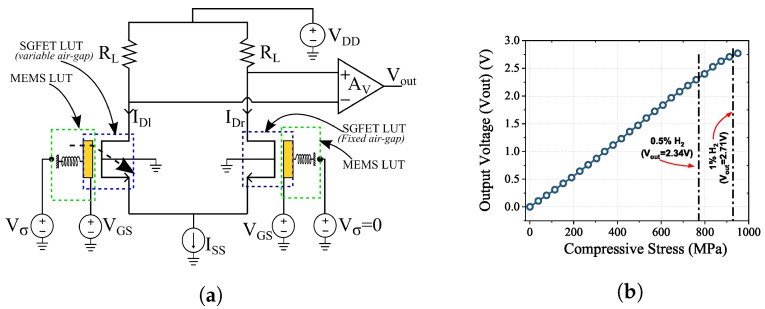
At $V_{GS}$ = 3 V, (**a**) the differential amplifier circuit of the RFM-SGFET hydrogen sensor, (**b**) output voltage plot of the differential amplifier circuit for different input stress.

**Figure 8 micromachines-14-00944-f008:**
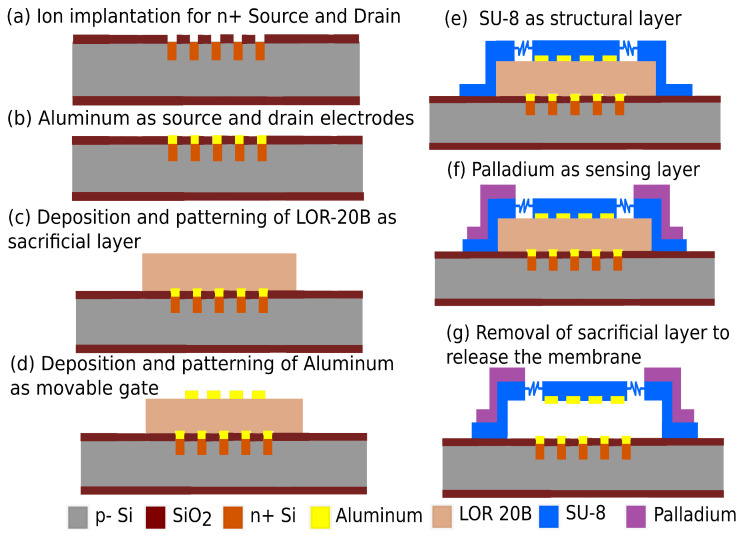
Fabrication process flow for RFM-SGFET sensor.

**Table 1 micromachines-14-00944-t001:** RFM-SGFET design parameters.

PARAMETERS	VALUES
Substrate doping ($N_{a}$)	1 × 10${}^{15}\;$cm${}^{-3}$
Arsenic ion-implantation dose	5 × 10${}^{14}$ cm${}^{-2}$
Ion implantation energy	40 keV
Gate oxide thickness ($t_{ox}$)	3 nm
Gate length (L)	10 μm
Gate width (W)	210 μm
Junction depth ($x_{j}$)	0.35 μm
Air gap ($t_{air}$)	2 μm

## Data Availability

The data supporting reported results can be made available via requesting the corresponding author.

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
