# Peer review of "Polymer Ring–Flexure–Membrane Suspended Gate FET Gas Sensor: Design, Modelling and Simulation"

_micromachines, 2023, doi:10.3390/mi14050944_

Round 1
Reviewer 1 Report
This work presents the detection of gas using ring-flexure-membrane (RFM) suspended gate field effect transistor (SGFET). Though the sensor characteristics is impressive, the paper presentation lacks many details. Of utmost importance, the sensor appears to be clearly sensitive to hydrogen; however, it has not been tested with other gases that may be absorbed by palladium, such as oxygen, nitrogen, carbon monoxide, carbon dioxide, or hydrogen sulfide. Additionally, the reversibility of this sensor has not been commented upon.
Additional questions:
1. Could the authors please provide the conditions for the electrical measurements, such as VDS, VGS, gas compositions, and so on? It would also be helpful to know how many sensors were tested and to have a detailed the errors of the measurements and simulations. The authors did not provide sufficient information on the testing conditions in the current manuscript.
2. Additionally, it would be beneficial to include some real images of the prototype in the manuscript.
From a scientific standpoint, I did not observe any new developments presented in this manuscript. The sensor fabrication process is an update of previously published works, and the device measurement and testing principles are well-known among experts in the field. Therefore, I would not recommend publishing this manuscript in its current format.
However, if the manuscript highlights the improvements made and provides greater selectivity and sensitivity, it may still have the potential to be published. In order to achieve this, a more detailed description of the fabrication and characterization process would be necessary. Therefore, I suggest that the authors revise and improve the manuscript accordingly.
Reviewer 2 Report
It is about a computational study and proposal of a Polymer Ring-Flexure-Membrane Suspended gate FET-based gas sensor. The Synopsis Sentaurus TCAD has been used as a simulator. The working principle is based on the gas-induced vertical displacement in nanogap height that modulates the device characteristics leading to a change in drain current. The manuscript can be improved. The reviewer's comments are below:
1) The authors have said that ''The working of the RFM-SGFET sensor is shown in Fig.1(b). During gas adsorption, compressive stress is induced in the sensing layer, which causes the deformation of the ring and flexure, leading to a downward displacement of the membrane. This gate displacement leads to an increase in gate capacitance of the SGFET, thus causing a change in the output drain current''. In which basis this statement and working principle is based? Is there any experimental results? I recommend to add reference regarding this statement on which all the work is based.
2) Throughout the manuscript i did not see any sensitivity versus gas concentration or gas pressure. The authors should elaborate the computational relation between the displacement and gas concentration and gas type (if this working principle is sound and have an experimental support). Else this is a pressure or Nano-force sensor.
3) Even if a cantilever system has been used in conjunction with the FET transducer. The inclination of the cantilever has not been considered in the simulation, while the authors have assumed a uniform decrease in nanogap height when the gas is exposed.
4) I recommend to study while citing these relevant work to note only enrich the introduction section but also to provide to the readers a kind of guidance for further computational investigations: DOI : 10.1038/ncomms14950 ; DOI: 10.1016/j.aeue.2020.153346
5) I recommend to proofread the manuscipt to improve the narration.
Good luck.
Round 2
Reviewer 1 Report
After reviewing your comments, I believe that the article could be published with some minor modifications to the title and abstract. Since the article is about a concept rather than a prototype, I suggest that the title and abstract be adjusted accordingly. To avoid confusion, the title should incorporate terms such as design, modeling, and simulation. This title will be reserve for the publication of the real device. Once this change has been made, the article can be published without issue.
Author Response
We would thank the reviewer for accepting our manuscript with minor revision. Based on the reviewers comment, we have revised the title and abstract of the manuscript.